# 2′-Fucosyllactose Suppresses Angiogenesis and Alleviates Toxic Effects of 5-Fu in a HCT116 Colon Tumor-Bearing Model

**DOI:** 10.3390/molecules27217255

**Published:** 2022-10-26

**Authors:** Huiying Li, Bingyuan Wang, Yang Wang

**Affiliations:** 1Beijing Key Laboratory of Food Processing and Safety in Forest, College of Biological Sciences and Technology, Beijing Forestry University, Beijing 100083, China; 2Institute of Animal Sciences, Chinese Academy of Agricultural Sciences, Beijing 100193, China; 3State Key Laboratory of Membrane Biology, Tsinghua University-Peking University Joint Center for Life Sciences, School of Life Sciences, Tsinghua University, Beijing 100084, China

**Keywords:** 2′-FL, 5-Fu, HCT116 cell, tumor-bearing model, angiogenesis

## Abstract

The present study was aimed at examining the anti-tumor effects and molecular mechanisms of 2′-fucosyllactose (2′-FL). At the beginning, the viabilities of four types of colon cancer cells were analyzed after exposure to increasing concentrations of 2′-FL, and HCT116 cells were selected as the sensitive ones, which were applied in the further experiments; then, interestingly, 2′-FL (102.35 µM) was found to induce apoptosis of HCT116 cells, which coincides with significant changes in VEGFA/VEGFR2/p-PI3K/p-Akt/cleaved Caspase3 proteins. Next, in a tumor-bearing nude mouse model, HCT116 was chosen as the sensitive cell line, and 5-fluorouracil (5-Fu) was chosen as the positive medicine. It was noteworthy that both 2′-FL group (2.41 ± 0.57 g) and 2′FL/5-Fu group (1.22 ± 0.35 g) had a significantly lower tumor weight compared with the control (3.87 ± 0.79 g), suggesting 2′-FL could inhibit colon cancer. Since 2′-FL reduced the number of new blood vessels and the malignancy of tumors, we confirmed that 2′-FL effectively inhibited HCT116 tumors, and its mechanism was achieved by regulating the VEGFA/VEGFR2/PI3K/Akt/Caspase3 pathway. Moreover, though HE staining and organ index measurement, 2′-FL was validated to alleviate toxic effects on liver and kidney tissue when combining with 5-Fu. In conclusion, 2′-FL had certain anti-tumor and detoxification effects.

## 1. Introduction

Colorectal cancer is one of the three cancers in the world with the highest mortality. Its pathogenic factors are closely related to age and lifestyle [1,2]. In 2012, the global cancer incidence rate and mortality survey found that among 1,360,000 newly diagnosed colorectal cancer cases, 694,000 cases died within five years [3]. In China, the incidence and mortality of colorectal cancer are higher than the world average levels over the past few years, for areas with rapid development of economy and urbanization, as well as areas known for unhealthy diet style and lacking of exercise [4]. Surgery and chemotherapy are the most common clinical treatments for colorectal cancer, and, unfortunately, almost all patients have postoperative complications, including adverse side-effects and poor quality of life [4,5]. In order to reduce these treatment complications, it is necessary to study a variety of candidate drugs which have less toxicity and higher efficacy for colorectal cancer. During its occurrence and development, colorectal cancer is often accompanied by abnormal regulation of cell proliferation, apoptosis, cell cycle arrest, and other related signal pathways [6]. For example, increasing studies have found that PI3K/Akt-related inflammatory pathways are dysregulated in colorectal cancer cells [7], therefore, anti-inflammatory, cytotoxic, pro-apoptotic, and cycle-regulating drugs are often applied to treat colorectal tumors [8,9]. At present, the conventional basic chemotherapy drug for the systematic treatment of colorectal cancer is 5-fluorouracil (5-Fu), which is always used after surgery [10]. Nevertheless, the majority of patients are over-sensitive to 5-Fu-based treatment, resulting in serious side-effects and even death [11]. Although 5-Fu is rapidly excreted from the body, a small portion of the drug is metabolized and a portion is accumulated in liver tissue [12]. More terrifying, 5-Fu was associated with multiple toxic effects, including neuropathy, leukocyte transplantation, cardiotoxicity, hepatotoxicity, and nephrotoxicity [13]. Therefore, it is of great importance that a natural component is found that can not only reduce the side-effects of cancer chemotherapy drugs, such as 5-Fu, but also enhance the efficacy of these drugs.

As an initial nutrition source, human milk contains various bioactive proteins, fats, micronutrients, and prebiotics, which contribute significantly to the growth and development of neonates during their first months. As reported, breastfed infants were not susceptible to inflammation and hypoimmunity mainly due to the presence of multiple oligosaccharides with different structural lengths in human milk, known as human milk oligosaccharides (HMOs), which were present in smaller amounts in milk powder [14]. Until now, more than 200 types of HMOs have been identified, mainly including fucosylated, sialylated, and neutral lactoses [15]. Due to the presence of fucosylated and sialylated radicals, the majority (about 98%) of HMOs cannot be digested by enzymes in the gastrointestinal tract, and, instead, can reach the gut microbiota and be fermented in the colon tissue [16]. It is the most abundant HMO in human milk, the portion of 2’-fucosyllactose (2′-FL) being approximately 25%. It had the effect of regulating the immune systems of newborns by increasing the abundance of intestinal probiotics [17]. 2′-FL was verified to suppress the adhesion of pathogens to the surface glycans of epithelial cells, thus inhibiting the virulence of several pathogens [14,18]. Previous studies indicated that 2′-FL could efficiently prevent necrotizing enterocolitis (NEC) and decrease high mortality in neonates [19,20]. 2′-FL was also proven to alleviate inflammatory injury through regulating the expression of CD14 [21] or reducing the concentration of cytokines in infant plasma [22]. Myriam evaluated the safety of 2′-FL in a young rat model and found that there were no obvious side-effects of young rats being treated with 2′-FL (5 g/kg body weight, gavage administration) successively for 90 days, which provided solid support for its addition in infant foods [23]. Based on a batch culture fermentation model and pilot clinical trial findings, Ryan proved that 2′-FL improved chronic gastrointestinal conditions through regulating gut microbiota composition in adults [24]. Zhao found that 2′-FL could ameliorate chemotherapy-induced intestinal mucositis by protecting intestinal epithelial cells against apoptosis [25]; therefore, in 2015 and 2016, it was approved by the Food and Drug Administration of the United States (FDA) to serve as a dietary supplement and nutritional health-care product, and the recent rapid progress of biotechnology and fermentation engineering could enable the gradual mass production of 2′-FL.

Referring to the anti-tumor research of 2′-FL, a few studies have verified that 2′-FL suppressed gastrointestinal tumors. Heo found that 2′-FL promoted melanin degradation via the autophagic AMPK–ULK1 signaling axis [26]. In Azagra-Boronat’s study, animals given 2′-FL showed higher plasma IgG and IgA and more T cell subsets in the mesenteric lymph nodes on day 16, indicating that 2′-FL might have anti-cancer effects through immunomodulatory and prebiotic pathways [27]; thus, in the present study, we evaluated the anti-tumor potentials of 2′-FL in a xenograft colon tumor model and explored the molecular mechanism on how 2′-FL exerted effects in inhibiting colon cancer. Furthermore, we investigated the role of 2′-FL in alleviating organ toxic effects when combined with 5-Fu, a classical chemotherapeutic drug of colon cancer. This study provided new insights in understanding the relationship between milk-derived nutrients and human health, as well as new approaches to treat colon tumors in the clinical field.

## 2. Results

### 2.1. Cells Viability Assay and Dosages Selection

As Figure 1B shows, high dosages of 2′-FL (20.47–1023.54 µM) inhibited cells’ viability in a dose-dependent manner, with the 50% concentration of inhibition (IC50) at 1747.94 ± 90.80 µM, 1304.48 ± 55.54 µM, 1081.13 ± 70.97 µM, and 1468.58 ± 80.29 µM for DLD1, HT29, HCT116, and SW620 cells, respectively. The IC50 of HCT116 cells was the smallest one and we regarded the HCT116 cell line to be the most sensitive one to the treatment of 2′-FL, when compared with the other three cell lines. Moreover, according to the effective dosage selection principle, with the treatment of 2′-FL at 102.35 µM, the HCT116 cells’ viability was below 80%, and there was significant difference of cell viability between the HCT116 group and the other groups (*p* < 0.05), thus, we selected HCT116 as the sensitive cell line, and selected 102.35 µM (50 mg/L) as the proper dosage of HCT116 cells for the following in vitro experiments.

### 2.2. The Apoptosis Effect of 5-Fu and 2′-FL on HCT116 Cells

Using 5-Fu as a positive control, the apoptosis effect of 2′-FL on HCT116 cells was observed. According to the apoptosis instrument stained with annexin V, the cells in the LR region (lower right quadrant) were early apoptotic cells, so we compared their LR ratios. The results showed that 5-Fu, 2′-FL, and 5-Fu + 2′-FL could induce apoptosis in different degrees of HCT116 cells. The FL combined treatment group (15.23% ± 1.94%) was compared with single 5-Fu (12.70% ± 1.51%) or single 2′-FL (7.25% ± 0.98%) (*p* < 0.05, Figure 2).

### 2.3. 2′-FL Suppressed Xenografted Colon Tumors in a Nude Mice Model

To investigate the anarei-tumor effects of 2′-FL in vivo, a xenografted tumor model with HCT116 cells was constructed, and 5-Fu was selected as the positive medicine to compare the anti-tumor effects of 2′-FL and 5-Fu. Results in Figure 3 show that the average tumor weights in four groups were 3.87 ± 0.66 g (control), 1.59 ± 0.25 g (5-Fu), 2.42 ± 0.55 g (2′-FL), and 1.21 ± 0.20 g (5-Fu + 2′-FL), respectively (Figure 3A). The tumor suppression rate in (5-Fu + 2′-FL) group was the highest one (*p* < 0.05, Figure 3B), and the relative tumor proliferation rate in (5-Fu + 2′-FL) group was the smallest one (*p* < 0.05, Figure 3C), when compared with the single 5-Fu or 2′-FL treatment groups. These results were consistent with the in vitro data that the co-treatment of 5-Fu and 2′-FL showed stronger anti-tumor effects, which might be a novel therapy in clinical cancer treatment area.

### 2.4. 2′-FL Reduced the Degrees of Tumor Malignancy and Angiogenesis

To explore further the related mechanism of 2′-FL in suppressing HCT116 tumors, we conducted HE staining and CD34 staining of tumor tissue, and found that the numbers of newborn blood vessels in tumors treated with 5-Fu, 2′-FL, or the co-treatment groups were less than the control level (Figure 4A), and the ones in the co-treatment group were the smallest (Figure 4B); besides, the edematous and hemorrhage areas in the treatment groups were also smaller when compared with the control group (Figure 4A).

### 2.5. 2′-FL Alleviated the Toxic Effects When Combined with 5-Fu

Next, we measured the organ indexes of mice to evaluate the toxicity of 5-Fu and 2′-FL to multiple organs and observed their immune function. It can be seen from Table 1 that the organ indexes of the 2′-FL group and the lung, liver, and kidney indexes of the 5-Fu group were higher than those of the control group (*p* < 0.05). The thymus index was significantly decreased (*p* < 0.05); the indexes of heart/liver/kidney in (5-Fu + 2′-FL) group were significantly higher than those in the normal group (*p* < 0.05), and the liver/kidney indexes were lower than those in (5-Fu + 2′-FL) group. 2′-FL group (*p* < 0.05). The spleen/thymus index of (5-Fu + 2′-FL) group was lower than that of the normal group (*p* < 0.05), and the spleen/thymus index was higher than that of the 5-Fu group (*p* < 0.05). The above results suggest that 5-Fu has caused certain damage to the body’s immune function, especially noting that 2′-FL has a certain protective effect on the toxicity of 5-Fu.

In addition, the pathological changes of the liver and kidney were observed by HE staining, and the results showed that the 2′-FL group and the combined group had no significant damage to the heart, liver, and kidney (see Figure 4A). Edema, changes in cell morphology, and even hemorrhage were also seen in the 5-Fu group, whereas in the co-treatment group, 2′-FL alleviated these lesions, which is consistent with the data on organ indicators (Figure 4A).

### 2.6. 2′-FL Regulates the Angiogenesis Pathway of VEGFA/VEGFR2 and Induces Apoptosis in HCT116 Cells

The VEGFA/VEGFR2/PI3K/Akt/Akt signal transduction method by VEGFRs was used to measure the VEGFA/VEGFR2/PI3K/Akt signal transduction pathway, thereby revealing the mechanism of 2′-FL on colorectal cancer. The expression of the protein showed that 2′-FL significantly reduced the levels of VEGFA/VEGFR2/pPI3K/pAkt proteins in HCT116 and DLD1 cells, and increased the levels of Caspase3 proteins; however, the contents of PI3K, Akt, Erk1, and Caspase3 did not change significantly, indicating that 2′-FL can reduce kinase phosphorylation and activate apoptosis (Figure 5). 2′-FL regulated the VEGFA/VEGFR2 angiogenesis pathway and induced apoptosis in HCT116 cells.

## 3. Discussion

The classical feature of angiogenesis is the accumulating formation of new blood vessels. It plays a key role in tumor growth and metastasis. Angiogenesis participates in transporting oxygen and nutrients for tumors, removing waste from tumor tissues, and providing passageways for tumor metastasis. Cutting off angiogenesis might suppress cancer development in some degree [28]. The activation of the vascular endothelial growth factors (VEGFs) pathway is pivotal for angiogenesis; the levels of VEGFs sharply increase and then angiogenesis is induced in tissue hypoxia [29]. As a well-studied factor, VEGFA acts through VEGF receptor 2 (VEGFR2) on endothelial cells (ECs). The latter can not only germinate in the existing vascular system, but also can reform blood vessels through ECs precursors and angioblasts [28]. Furthermore, the combination of VEGFA and VEGFR2 activates multiple signaling pathways, including PI3K/Akt or MAPK, which induce endothelial cell proliferation, survival, adhesion, and migration, and form new blood vessels [30]. Detection of the expression levels of VEGFA and VEGFR2 in blood samples of patients with metastatic colorectal cancer are a direct and accurate method to monitor the effects of anti-angiogenesis therapy [31]. In general, without angiogenesis, cancer cannot grow and develop, and threaten human life; therefore, it is of great significance to study the drugs that inhibit angiogenesis in clinical treatments of cancers. At present, several clinical anti-angiogenesis drugs, such as 5-Fu, have great toxicity and adverse side-effects [32]. On the whole, the toxicities of components from natural medicinal plants and food products have relatively tiny impact on the human body, and, if natural ingredients or food nutrients can be proven to alleviate toxicities of chemotherapeutic drugs and enhance anti-tumor effects when combining with chemotherapeutic drugs, they will play an important and positive role in the treatment of cancer.

In cell viability assay, we found 2′-FL with 20.47–1023.54 µM could inhibit cells’ survival in four types of colorectal cancer cells, and, especially, the HCT116 cell line was selected as the sensitive one for its IC50 was the smallest. Next, animal experiment results demonstrated that 2′-FL could suppress HCT116 xenografted tumors, and the suppressive effect of the 5-Fu + 2′-FL group was the strongest, when compared with the single 5-Fu or 2′-FL group. Considering that there appeared to be no study reporting the anti-tumor effects of 2′-FL, the present results in vitro and in vivo for the first time proved the functions of 2′-FL in inhibiting HCT116 tumors, as well as the possible enhancement of 5-Fu + 2′-FL.

The effect of 2′-FL in HCT116 cells’ apoptosis was also investigated, suggesting that the co-treatment of 5-Fu and 2′-FL, indeed, showed a stronger effect in inducing apoptosis of colon cancer cells, when compared with the single-treatment groups. The results of this part of the research suggested the possible mechanism of 2′-FL in inhibiting HCT116 tumors, which would be verified later by Western blotting.

In order to explore the mechanism of 2′-FL on colorectal cancer, HE, CD34, and other methods were used. The results showed that after 5-Fu and 2′-FL treatment, the number of blood vessels in the tumor was the least. In this paper, the tumor tissue sections after 2′-FL treatment were analyzed, and the effect of 2′-FL on VEGFA/VEGFR2/PI3K/Akt was investigated. 2′-FL decreased VEGFR2, VEGFA, and pPI3K in a dose-dependent manner; the results showed that 2′-FL was active in VEGFA/VEGFR2/PI3K/Akt/Akt signal transduction pathway and activation of apoptotic factors. In the future, the binding information of 2′-FL and VEGFA/VEGFR2 protein will be confirmed, and candidate targets of 2′-FL can be identified by over-expression or small interfering RNA (siRNA); especially, the direct effect of 2′-FL on angiogenesis will be further investigated by experiments, including cell proliferation, cell cycle, and cell migration, as well as tube formation detection of vascular endothelial cells (such as human umbilical vein endothelial cells), which are significant to elucidating further the anti-tumor mechanisms of 2′-FL.

More interestingly, we found that 2′-FL alleviated the toxicity of 5-Fu, mainly improving organ pathological conditions, down-regulating organ indices, and strengthening immunizing power. Based on the possible enhancement of 5-Fu and 2′-FL in suppressing HCT116 cells’ viability and HCT116 tumors’ growth, we proved that 2′-FL exerted anti-tumor and toxicity alleviation effects in the present study.

## 4. Materials and Methods

### 4.1. Chemicals

DLD1, HT29, HCT116, and SW620 cell lines were purchased from Shanghai, China. Roswell Park Memorial Institute (RPMI) 1640 medium, fetal bovine serum (FBS), and 100× (100×) were acquired from Gibco (New York), New York, NY, USA. 2′-FL of more than 98% purity was purchased from Huichi Technology (Beijing, Shanghai). A CCK-8 kit, an apoptosis-detection kit, formaldehyde solution, and HE and IHC staining kits (Shanghai, China) were purchased from Beyotime (Beyotime). VEGFR2, VEGFA, pPI3K; PI3K, pAkt, Akt; clevaged Caspase3, Caspase3, CD34, and β-actin primary antibodies and their secondary antibodies were available from Santa Cruz, CA, USA. Other reagents for Western blotting and enhanced chemiluminescence (ECL) were purchased from Solarbio (Beijing, China). 5-Fu with a purity of 98% was purchased from MCE (Shanghai, China).

### 4.2. Cell Culture and Activity Detection

The cells were incubated with 10% FBS, 1% penicillin/streptomycin in RPMI-1640 at 37 °C with 5% CO_2_. These cells can be delivered in about 3 days. Four kinds of cells (1 × 104 cells, 200 μL per well) were seeded in 96-well plates, cultured for 24 h, and then treated with different concentrations of 2′-FL containing 0–1023.54 µM. Next, according to the protocol, 10 microliters of CCK-8 solution was co-incubated for 4 h, and then cell viability was determined using an enzyme labeler (Simoft, California, USA) under a light intensity of 490 nm (A). Cell viability = (A test–A blank) × 100%. Then, in the following in vitro and in vivo experiments, a sensitive cell line (HCT116) was selected.

### 4.3. Apoptosis Detection

By studying the apoptosis of HCT116 cells, the effects of different treatment conditions on the apoptosis of HCT116 cells were discussed. Briefly, cells were grown in 6-well plates and treated with 5-Fu (2.29 μM), 2′-FL (102.35 μM), and (5-Fu + 2‘-FL) for 48 h. After HCT116 cells were washed in PBS buffer, HCT116 cells were incubated in 200 μL of binding buffer according to the method of Annexin V-FITC (Solarbio, China) and V-FITC (Solarbio, China). They were suspended in the solution. AnnexinV-FITC and PI buffer (10, 20 g/L) were added to the cell suspension, respectively, and incubated in the dark (25 °C) for 10 min, followed by adding 300 μL of binding agent to each test tube. A buffer process followed. Samples were measured by flow cytometry for 90 min. Finally, cells in AV+/PI+ (lower right quadrant) were considered necrotic and late apoptotic, AV−/PI− (lower right quadrant) were early, AV− /PI− was active, while AV+ and PI− were in apoptosis.

### 4.4. Animal Tests

To verify the inhibitory effect of 2′-FL on colorectal cancer, the HCT116 nude mouse xenograft model was established. Large-scale HCT116 cells (10, 10, 10 cm in diameter) were cultured and injected into 20 (18 to 22 g) BALB/c nude mice, from Beijing Weitehe Experimental Biotechnology Co. Ltd. (Beijing, China). Next, 1 × 10^7^ cells (1 dish) were injected subcutaneously into the right side of the nude mice in 250 microliters of matrigel (Corning). When the tumor size was 90~110 mm^3^, the nude mice were randomly divided into four groups (of five mice each), with a control group (untreated), 5-Fu (3.81 µM/kg b.w), 2′-FL (409.42 µM/kg b.w, via gavage), and 5-Fu + 2′-FL (combined with 5-Fu and 2′-FL). 5-Fu, 2′-FL, or (5-Fu + 2′-FL) was administered every 2 days. On day 25, all mice were killed, and the tumors and several organs were dissected, including heart, liver, kidney, spleen, and thymus. The animal experiments were reviewed by the Ethics Committee of the Chinese Academy of Agricultural Sciences (Beijing, China), and an IAS-2021-03 approval was obtained and permission granted.

Porportion of tumor suppressor = (tumor weight in control group−protein treatment group)/tumor mass in control group × 100%;

Relative volume of tumor (RTV, %) = drug dose / volume before dose ∗ 100%;

Relative proliferation rate of tumor = RTV/RTV in control group/RTV in protein group × 100%;

Organ index = organ mass/body mass × 100%.

### 4.5. HE and IHC Methods

In order to study the pathological changes of 2′-FL on tumor, liver, and kidney, CD34 was detected by hematoxylin and eosin (HE) and immunohistochemistry (IHC). HE-stained tissue sections were observed with a light microscope, and CD34-stained sections were collected with a confocal laser scanning microscope (Olympus, Tokyo, Japan).

### 4.6. Western Blotting

In order to explore the mechanism of 2′-FL in colorectal cancer, this study used Western blotting technology to study its related factors. All proteins in HCT116 cells were lysed with a lysing agent (Solarbio) containing phosphatase and protease inhibitors, followed by centrifugation at 8000× *g* for 15 min at 4 °C and 15 min at 100 °C. The protein samples were then loaded on a 12% SDS-polyacrylamide gel, transferred to a nitrocellulose filter by a Trans-Blot machine (Thermo), and buffered with 5% BSA in TBST within 25 min. The film was blocked for 1 h. At 25 °C, the protein was detected with the primary antibody for 2.5 h, and β-actin was used as the internal reference to ensure the quality of the sample. The membrane was then washed with PBST buffer (3 × 5 min), then incubated with the secondary antibody for 1.5 h at 25 °C, and then rinsed (10 min × 3). Finally, the signal on the film was detected with ECL reagent and analyzed with Image J software.

### 4.7. Data Analysis

All data are expressed as mean–standard deviation (SD) and analyzed using SPSS 19.0 and GraphPad 6.0 (GraphPad Inc., San Diego, CA, USA). Student’s t-test was used to compare the differences between the two groups (control group and treatment group, 5-Fu + 2′-FL group). A *p*-value < 0.05 was considered statistically significant.

## 5. Conclusions

To conclude, this study’s results showed that 2′-FL showed obvious anti-tumor effect both in vitro and in vivo. When combined with 5-Fu, 2′-FL could effectively reduce its toxicity and adverse reactions. In addition, 2′-FL could also be specifically sensitive to a certain type of colon cancer cell, such as HCT116, and its mechanism was proved to be related to the VEGFA/VEGFR2/PI3K/Akt pathway, and it could also induce downstream apoptosis factors. However, the effectiveness of 2′-FL in clinical trials is still unclear and requires multiple experiments involving such factors as clinical resources, colon cancer patient volunteers, funding, etc., which could not be met by current biological laboratories. With further pharmacokinetic and toxicological studies data, 2′-FL, or its application in combination with other chemotherapies (e.g., paclitaxel or 5-Fu), could be considered as alternative treatments.

## Figures and Tables

**Figure 1 molecules-27-07255-f001:**
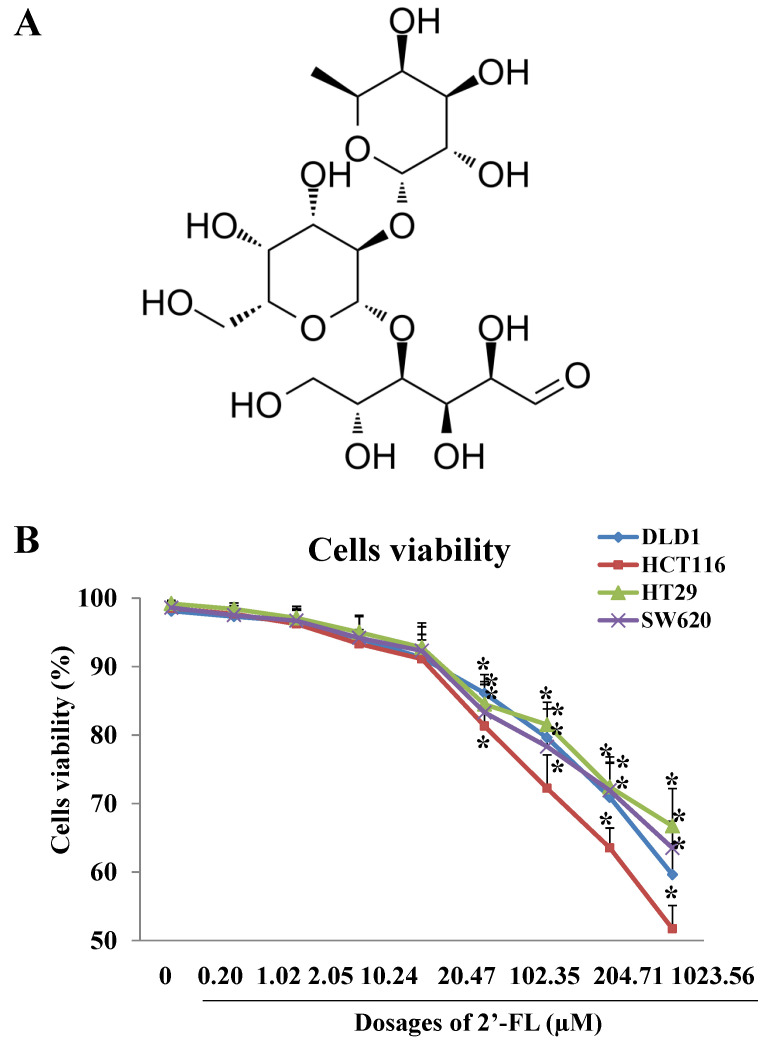
The effects of 2′-FL on cell survival rate in DLD1, HCT116, HT29, and SW620 cells models. (**A**) Chemical structure of 2′-FL. (**B**) DLD1, HCT116, HT29, and SW620 cells survival rates affected by 2′-FL with different concentrations. All data in (**B**) are presented as mean ± SD, * *p* < 0.05 compared with the control group (0 mg/L), *n* = 4.

**Figure 2 molecules-27-07255-f002:**
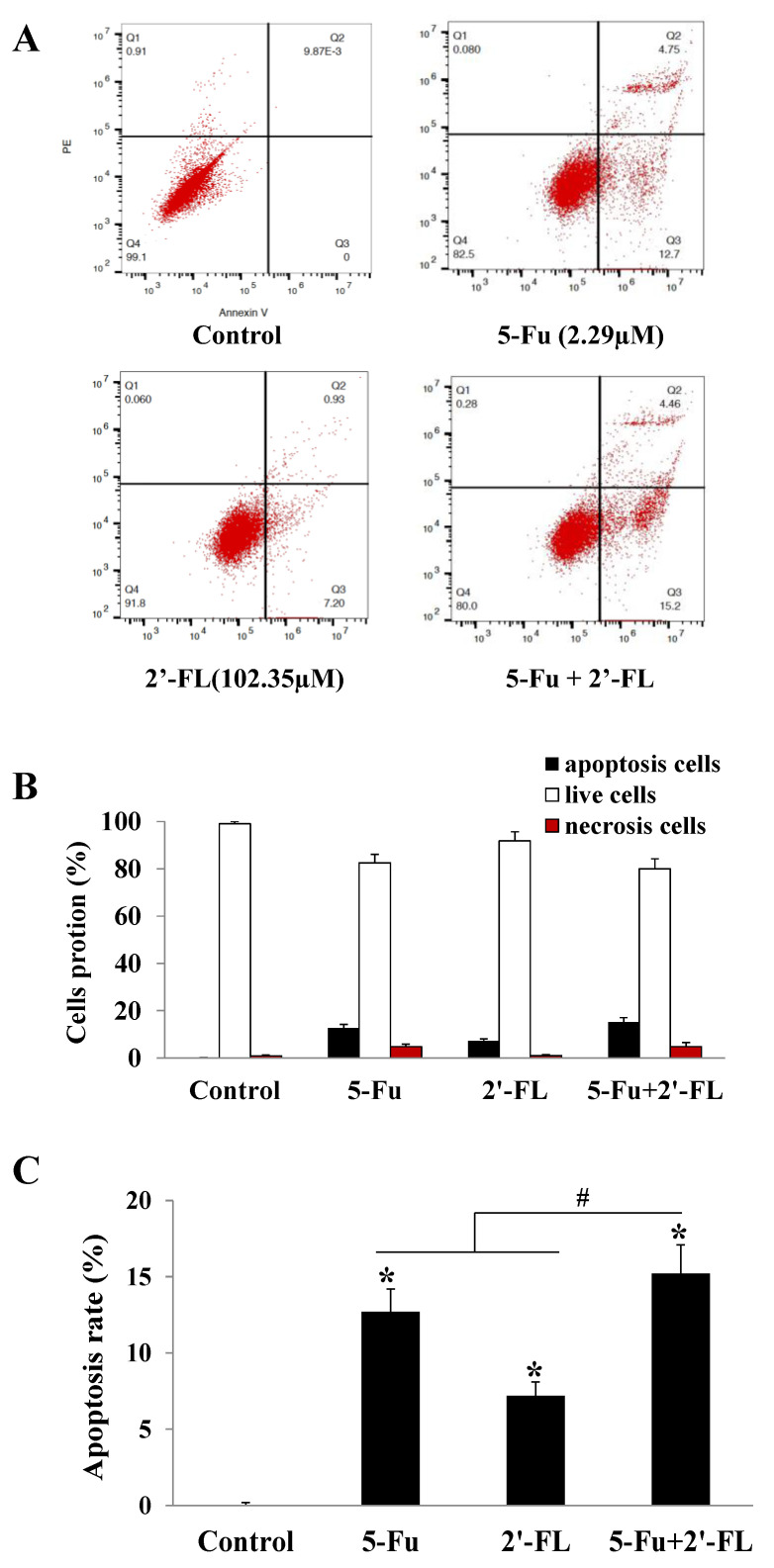
The effect of 2′-FL (102.35 µM) on HCT116 cells apoptosis. (**A**) HCT116 cells apoptosis affected by 5-Fu (2.29 µM) and 2′-FL (102.35 µM). (**B**) Cells’ portions of apoptosis, necrosis’ and live ones detected by FACS. (**C**) The cells apoptotic rates of HCT116 cells treated with 5-Fu, 2′-FL’ or (5-Fu + 2′-FL). The data are presented as mean ± SD, * *p* < 0.05 compared with the control group, ^#^ *p* < 0.05 compared with (5-Fu + 2′-FL) treatment group, *n* = 3.

**Figure 3 molecules-27-07255-f003:**
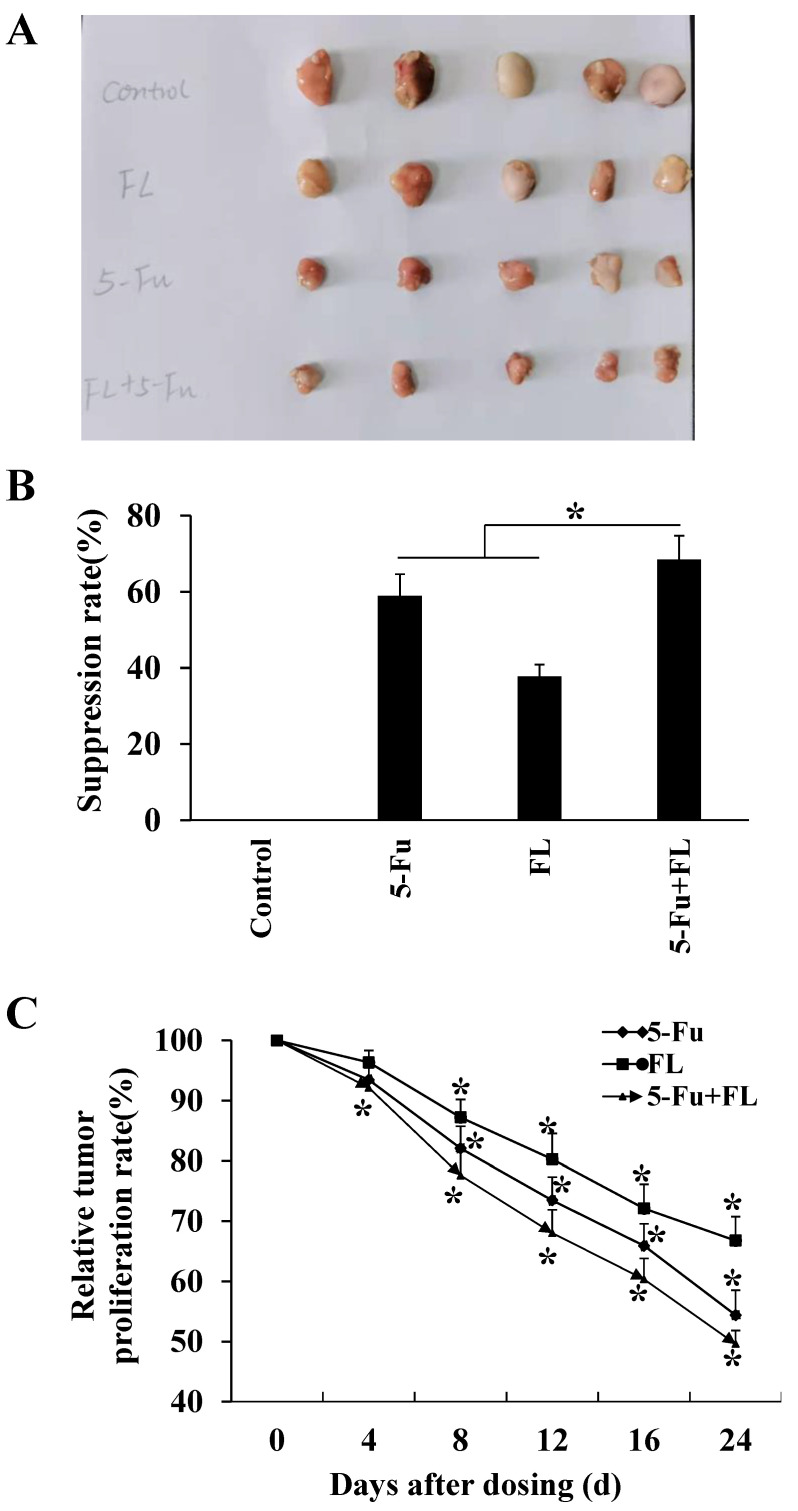
2′-FL-inhibited HCT116 xenografted tumors in the BALB/c nude mouse model. (**A**) The comparison of tumors in different groups, including control without any treatment, 5-Fu group (3.81 µM/kg b.w.), 2′-FL group (409.42 µM/kg b.w.), and (5-Fu + 2′-FL) group. (**B**) The tumor weights in BALB/c nude mouse model, which were treated with 5-Fu group (3.81 µM/kg b.w.), 2′-FL group (409.42 µM/kg b.w.), and (5-Fu + 2′-FL), respectively. (**C**) The relative tumor volumes (RTVs) of tumors treated with 5-Fu (3.81 µM/kg b.w.), 2′-FL group (409.42 µM/kg b.w.), and (5-Fu + 2′-FL). The data are presented as mean ± SD, * *p* < 0.05 compared with the control group, *n* = 5.

**Figure 4 molecules-27-07255-f004:**
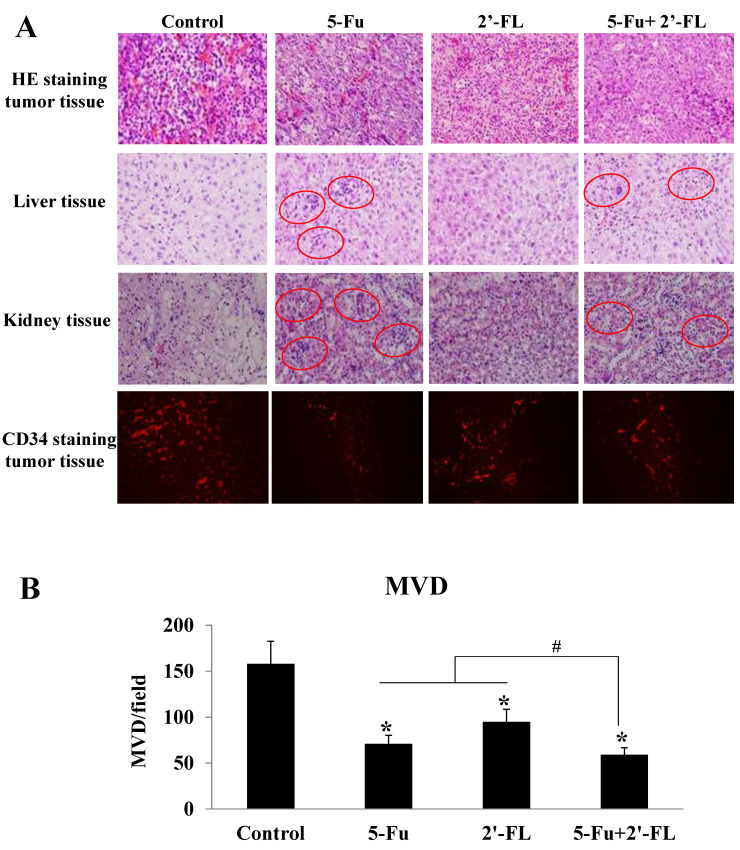
The effect of 2′-FL on HCT116 tumor angiogenesis detected by CD34 staining. (**A**) HE staining of HCT116 tumor tissue, liver tissue, and kidney tissue, as well as CD34 staining of HCT116 tumor tissue. (**B**) The microvessel density (MVD) of HCT116 tumor tissue by CD34 staining. The data are presented as mean ± SD, * *p* < 0.05 compared with the control group, and ^#^
*p* < 0.05 compared with (5-Fu + 2′-FL) group, *n* = 3.

**Figure 5 molecules-27-07255-f005:**
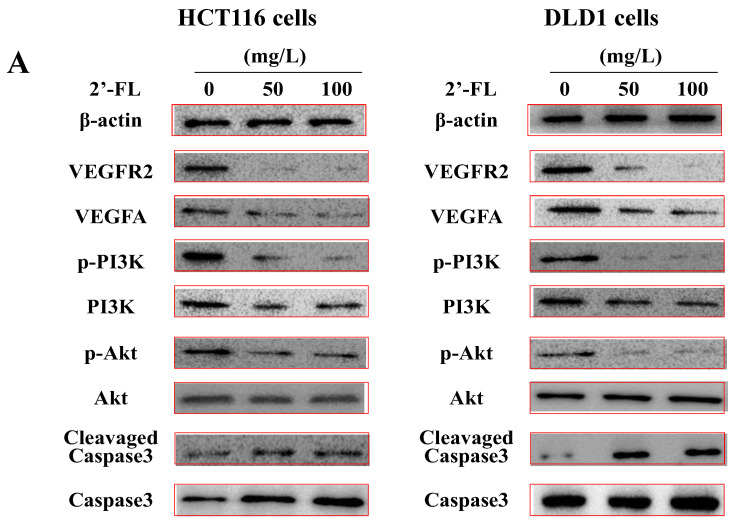
The protein levels of the angiogenesis-related pathway in HCT116 cells and DLD1 cells. (**A**) Western blot bands of the proteins of VEGFA/VEGFR2/p-PI3K/PI3K/p-Akt/Akt/cleavaged Caspase-3/Caspase-3 pathway. (**B**) Statistical data of the proteins expression levels in HCT116 cells. (**C**) Statistical data of the proteins’ expression levels in DLD1 cells. All data are presented as mean ± SD, * *p* < 0.05 compared with the normal group, *n* = 3.

**Table 1 molecules-27-07255-t001:** Mice organ injury index (mg/g).

Group (mg/Kg)	Heart Index	Liver Index	Kidney Index	Spleen Index	Thymus Index
Control	0.55 ± 0.06	6.02 ± 0.29	1.55 ± 0.20	0.51 ± 0.04	0.065 ± 0.007
5-Fu(0.5)	0.67 ± 0.07 *	6.85 ± 0.41 *	2.03 ± 0.37 *	0.35 ± 0.07 *	0.042 ± 0.009 *
2′-FL(100)	0.56 ± 0.04 ^#^	6.05 ± 0.33 ^#^	1.52 ± 0.17 ^#^	0.52 ± 0.07 ^#^	0.063 ± 0.010 ^#^
2′-FL + 5-Fu	0.63 ± 0.05 *	6.43 ± 0.45 *^,#^	1.68 ± 0.33 *^,#^	0.44 ± 0.09 *^,#^	0.052 ± 0.009 *^,#^

All the data were represented as mean ± SD (*n* = 5). Compared with control, * *p* < 0.05, compared with 5-Fu, ^#^ *p* < 0.

## Data Availability

Not applicable.

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
