# Peer review of "2′-Fucosyllactose Suppresses Angiogenesis and Alleviates Toxic Effects of 5-Fu in a HCT116 Colon Tumor-Bearing Model"

_molecules, 2022, doi:10.3390/molecules27217255_

Round 1
Reviewer 1 Report
In the present work, Li et al. studied the effect of 2'-fucosyllactose on colon cancer and found it can inhibit the growth of the HCT116 tumor in a xenograft mice model. As a polyphenol compound, it is no surprise that 2'-fucosyllactose showed an anti-tumor effect. The novelty of this manuscript is not significant from my point of view. Two key findings, suppressing angiogenesis and alleviating the toxic effect of 5-Fu, were not well supported by present data.
Author Response
In the present work, Li et al. studied the effect of 2'-fucosyllactose on colon cancer and found it can inhibit the growth of the HCT116 tumor in a xenograft mice model. As a polyphenol compound, it is no surprise that 2'-fucosyllactose showed an anti-tumor effect. The novelty of this manuscript is not significant from my point of view. Two key findings, suppressing angiogenesis and alleviating the toxic effect of 5-Fu, were not well supported by present data.
Answer: Thank you so much for your suggestions. We have revised the whole text and emphasized the key points of the present study more, which might be helpful to comprehend this manuscript.
Thank you very much for your meaningful suggestions and help!
Reviewer 2 Report
This manuscript deals with the important topic of the search for compounds that enhance the effectiveness of anti-cancer drugs, and at the same time can reduce their toxicity. I have some comments and questions:
It is not clear why the 50 mg /L of 2’-Fl was selected for further analysis.
In my opinion, in figure 3, the legend does not match the one in the diagram.
Authors used phrase: “combination group (5-Fu+ 2’Fl)” but it is a combination of drugs.
Why do authors give the angiogenesis pathway in the following order:VEGFR2/VEGF A? The first there is VEGF A that act via receptor – VEGFR2.
In figure 5 B there is result for VEGF A, in figure 5C there is result for VEGF C. Do Authors analysed different VEGF in different cell lines?
135 line: all 134 mice were killed He died – I think that it should be without He died
176 line: invitro – it should be the Italic type and with space: in vitro
182 line: The apoptosis effect of 25-Fu and 2'-FL on HCT116 cells – I think that 5-Fu should be.
Author Response
To reviewer 2
This manuscript deals with the important topic of the search for compounds that enhance the effectiveness of anti-cancer drugs, and at the same time can reduce their toxicity. I have some comments and questions:
- It is not clear why the 50 mg /L of 2’-Fl was selected for further analysis.
Answer: We must apologize for the missing information of cell line selection and dosage selection, and we have added these information in 3.1 section, as “For the IC50 of HCT116 cells was the smallest one, we regarded HCT116 cell linenwas the most sensitive one to the treatment of 2'-FL, when compared with the other three cell lines. Moreover, according to the effective dosage selection principle, with the treatment of 2'-FL at 50 mg/L, the HCT116 cells viability was below 80%, and there was significant difference of cell viabilty between HCT116 group and the other groups (P < 0.05), thus, we selected HCT116 as the sensitive cell line, and selected 50 mg/L as the proper dosage of HCT116 cells for the following invitro experiments.”
- In my opinion, in figure 3, the legend does not match the one in the diagram.
Answer: We have checked the legend of Figure 3 and find information of 3B is missing, and we have added it accordingly, as “2’-FL inhibited HCT116 xenografted tumors in the BALB/c nude mouse model. (A) The comparison of tumors in different groups, including control without any treatment, 5-Fu group (0.5 mg/kg b.w.), 2’-FL group (200 mg/kg b.w.), and (5-Fu+2’-FL) group. (B) The tumor weights in BALB/c nude mouse model, which were treated with 5-Fu group (0.5 mg/kg b.w.), 2’-FL group (200 mg/kg b.w.), and (5-Fu+2’-FL), respectively. (C) The relative tumor volumes (RTVs) of tumors treated with 5-Fu (0.5 mg/kg b.w.), 2’-FL group (200 mg/kg b.w.), and (5-Fu+2’-FL). The data are presented as mean ± SD, * p < 0.05 compared with the control group, n = 5.”
- Authors used phrase: “combination group (5-Fu+ 2’Fl)” but it is a combination of drugs.
Answer: Yes, we have discussed this point, and we have revised the phrase “combination group” to “(5-Fu+2’-FL) treatment group” all through the text. Thank you for your meaningful suggestion!
- Why do authors give the angiogenesis pathway in the following order:VEGFR2/VEGF A? The first there is VEGF A that act via receptor – VEGFR2.
Answer: Yes, we have corrected the inappropriate expression of “VEGFR2/VEGF A related pathway” to “VEGF A/VEGFR2” in the whole manuscript.
- In figure 5 B there is result for VEGF A, in figure 5C there is result for VEGF C. Do Authors analysed different VEGF in different cell lines?
Answer: We must apologize for our mistakes, and we have revised the information in Figure 5C as “VEGFA/VEGFR2 related pathway”.
Statistical data of the proteins expression levels in DLD1 cells. All data are presented as mean ± SD, * p < 0.05 compared with the normal group, n = 3.
- 135 line: all 134 mice were killed He died – I think that it should be without He died
Answer: Sorry, we have revised this sentence as “On day 25, all mice were killed, and the tumors and several organs were dissected”.
- 176 line: invitro – it should be the Italic type and with space: in vitro
Answer: Yes, we have corrected this word to “in vitro”.
- 182 line: The apoptosis effect of 25-Fu and 2'-FL on HCT116 cells – I think that 5-Fu should be.
Answer: Yes, we have reivsed the wrong word as “5-Fu” here.
To sum up, thank you very much for your meaningful suggestions and help!

Reviewer 3 Report
This is a well-written manuscript on the anti-tumour effects and molecular mechanisms of 2'-fucosyllactose HCT116 colon cancer cells. The experimental design is rigorous, and the manuscript is organized and easy for readers to follow and catch the information. After reading the document carefully, I believe that the authors can address the following issues to strengthen the manuscript:
1. In the introduction, the authors should explain some examples of 2'-fucosyllactose anticancer activity, including its anticancer activity on gastrointestinal tumours. Moreover, it would be interesting to include the following references:
Heo, H., Cha, B., Jang, D. et al. Human milk oligosaccharide 2'-fucosyllactose promotes melanin degradation via the autophagic AMPK–ULK1 signaling axis. Sci Rep 12, 13983 (2022). https://doi.org/10.1038/s41598-022-17896-4
Azagra-Boronat, M. Massot-Cladera, J. Mayneris-Perxachs, K. Knipping, et al. Immunomodulatory and Prebiotic Effects of 2'-Fucosyllactose in Suckling Rats. Front. Immunol., 2019, 10, 1773. doi: 10.3389/fimmu.2019.01773.
J.J. Ryan, A. Monteagudo-Mera, N. Contractor, G.R. Gibson. Impact of 2’-Fucosyllactose on Gut Microbiota Composition in Adults with Chronic Gastrointestinal Conditions: Batch Culture Fermentation Model and Pilot Clinical Trial Findings. Nutrients 2021, 13, 938. https://doi.org/10.3390/nu13030938
G. Zhao, J. Williams, M. K. Washington, Y. Yang, et al. 2'-Fucosyllactose Ameliorates Chemotherapy-Induced Intestinal Mucositis by Protecting Intestinal Epithelial Cells Against Apoptosis Cell. Mol. Gastroenterol. Hepatol., 2022, 13, 441-457. doi: 10.1016/j.jcmgh.2021.09.015.
2. Authors should convert concentration units from mg/L to mM
3. Check line 182: “3.2. The apoptosis effect of 25-Fu and 2'-FL on HCT116 cell”.
4. Why the HCT116 cell line is more sensitive than the other three?
Author Response
To reviewer 3
This is a well-written manuscript on the anti-tumour effects and molecular mechanisms of 2'-fucosyllactose HCT116 colon cancer cells. The experimental design is rigorous, and the manuscript is organized and easy for readers to follow and catch the information. After reading the document carefully, I believe that the authors can address the following issues to strengthen the manuscript:
- In the introduction, the authors should explain some examples of 2'-fucosyllactose anticancer activity, including its anticancer activity on gastrointestinal tumours. Moreover, it would be interesting to include the following references:
Heo, H.; Cha, B.; Jang, D.; Park, C.; Park, G.; Kwak,B.; Bin, B. Human milk oligosaccharide 2'-fucosyllactose promotes melanin degradation via the autophagic AMPK–ULK1 signaling axis. Sci Rep 12, 13983 (2022). https://doi.org/10.1038/s41598-022-17896-4
Azagra-Boronat, I.; Massot-Cladera, M., Mayneris-Perxachs, J.; Knipping, K; Pérez-Cano, F.J. Immunomodulatory and Prebiotic Effects of 2'-Fucosyllactose in Suckling Rats. Front. Immunol., 2019, 10, 1773. doi: 10.3389/fimmu.2019.01773.
J.J. Ryan, A. Monteagudo-Mera, N. Contractor, G.R. Gibson. Impact of 2’-Fucosyllactose on Gut Microbiota Composition in Adults with Chronic Gastrointestinal Conditions: Batch Culture Fermentation Model and Pilot Clinical Trial Findings. Nutrients 2021, 13, 938. https://doi.org/10.3390/nu13030938
Zhao, G.; Williams, J.; Washington, M.K.; Yang, Y.; Long, J.R.; Townsend, S.D.; Yan, F. 2'-Fucosyllactose Ameliorates Chemotherapy-Induced Intestinal Mucositis by Protecting Intestinal Epithelial Cells Against Apoptosis Cell. Mol. Gastroenterol. Hepatol., 2022, 13, 441-457. doi: 10.1016/j.jcmgh.2021.09.015.
Answer: Thank you so much for your meaningful suggestion! We have read these references and cited them into the Introduction part (as Ref 25-28), then described the anticancer activities of 2’-FL on different aspects, especially on gastrointestinal tumours.
- Authors should convert concentration units from mg/L to mM.
Answer: Yes, we have converted the concentration units from mg/L to mM or µM in the whole text.
- Check line 182: “3.2. The apoptosis effect of 25-Fu and 2'-FL on HCT116 cell”.
Answer: Sorry. We have corrected the wrong expression of “25-Fu” to “5-Fu” here.
- Why the HCT116 cell line is more sensitive than the other three?
Answer: It’s a good question, and we also wonder why HCT116 cells are more sensitive to the effect of 2’-FL, when compared with the other cell lines. Firstly, through taking literature survey, we found that HCT116 cells were firstly seperated from a man with colon cancer in 1979, however, both DLD1 cells and SW620 cells were seperated from colorectal cancer patients. In the present study, we found that HCT11 cells were more sensitive to 2’-FL when compared with DLD1 and SW620 cells in cell viability detection, which might be attributed to the different origins. Secondly, another reason might be the cancer metastasis abilities, in clinical area, the overall recovery rate of colon cancer patients is usually higher than the one of colorectal cancer patients, for colorectal cancer always show stronger metastasis ability than colon cancer, which is not easy to cure. The above two aspects might explain why HCT116 cells showed more sensitivity to the treatment of 2’FL.
To sum up, thank you very much for your meaningful suggestions and help!
Round 2
Reviewer 1 Report
The revised version did not significantly improve compared to the original one. As I mentioned last time, I am concerned that the supportive evidence for the 2'-fucosyllactose effect on angiogenesis is not enough. It is a great pity that the authors did not apply any further experiments to prove that. I'm not surprised that 2'-fucosyllactose possesses anti-tumor potential. However, the novelty is not significant.
Author Response
Comments: The revised version did not significantly improve compared to the original one. As I mentioned last time, I am concerned that the supportive evidence for the 2'-fucosyllactose effect on angiogenesis is not enough. It is a great pity that the authors did not apply any further experiments to prove that. I'm not surprised that 2'-fucosyllactose possesses anti-tumor potential. However, the novelty is not significant.
Answer: Dear reviewer, we must apologize for our unsatisfactory response to your suggestion. We have searched several articles to find the supportive evidence for the inhibition of 2’-FL on tumor angiogenesis, and added the description of anti-tumor effects of 2’-FL on different tumors into the “Introduction” section. However, there is very few paper proving 2’-FL inhibit colon tumor through suppressing angiogenesis. Thus, in the near future, we would like to investigate the effects of 2’-FL on angiogenesis, including detecting the proliferation, migration ability, invasive ability of HUVEC cells, as well as performing tube formation assay of HUVEC cells or SVEC4-10 cells. If possible, we anticipate submiting another manuscript regarding the further experimental results of 2’-FL to this journal. Thank you so much for your meaningful suggestions and consideration!